# Comparison between Slow Freezing and Vitrification for Human Ovarian Tissue Cryopreservation and Xenotransplantation

**DOI:** 10.3390/ijms20133346

**Published:** 2019-07-08

**Authors:** Sanghoon Lee, Ki-Jin Ryu, Boram Kim, Dahyeon Kang, Yoon Young Kim, Tak Kim

**Affiliations:** 1Department of Obstetrics and Gynecology, Korea University College of Medicine, 73, Inchon-ro, Seongbuk-gu, Seoul 02841, Korea; 2Department of Obstetrics and Gynecology, Seoul National University Hospital, Seoul 03080, Korea

**Keywords:** slow freezing, vitrification, ovarian tissue cryopreservation, xenotransplantation, fertility preservation

## Abstract

Two methods for the cryopreservation of human ovarian tissue were compared using a xenotransplantation model to establish a safe and effective cryopreservation method. Ovarian tissues were obtained from women who underwent benign ovarian surgery in the gynecology research unit of a university hospital. The tissues were transplanted into 112 ovariectomized female severe combined immunodeficient mice 4 weeks after slow freezing or vitrification cryopreservation. Tissues were retrieved 4 weeks later. Primordial follicular counts decreased after cryopreservation and xenotransplantation, and were significantly higher in the slow freezing group than in the vitrification group (*p* < 0.001). Immunohistochemistry and TUNEL assay showed that the Ki-67 and CD31 markers of follicular proliferation and angiogenesis were higher in the slow freezing group (*p* < 0.001 and *p* = 0.006, respectively) and DNA damage was greater in the vitrification group (*p* < 0.001). Western blotting showed that vitrification increased cellular apoptosis. Anti-Müllerian hormone expression was low in transplanted samples subjected to both cryopreservation techniques. Electron microscopy revealed primordial follicle deformation in the vitrification group. Slow freezing for ovarian tissue cryopreservation is superior to vitrification in terms of follicle survival and growth after xenotransplantation. These results will be useful for fertility preservation in female cancer patients.

## 1. Introduction

Fertility preservation in reproductive women undergoing cancer treatments is a major worldwide concern. Advancements in cancer treatment, including systemic chemotherapy and radiotherapy, have improved the curative rates of malignancies, but may significantly induce gonadal damage in young women [1,2]. Ovarian failure induced by cancer treatment is associated with infertility, endocrine dysfunction, menopausal symptoms, osteoporosis, low self-esteem, and decreased quality of life (QoL) in female cancer survivors [3,4]. Since the likelihood of assisted reproduction success is reduced after cancer treatment, early referral to fertility preservation is important for these patients [5,6,7].

Oocyte cryopreservation was recently accepted as a standard method for fertility preservation and is used worldwide [8,9]. Both oocyte and embryo cryopreservation require ovarian stimulation for approximately 2 weeks, which may result in delayed cancer treatment and excessive levels of estrogen exposure during stimulation [10]. For prepubescent girls and women that cannot undergo ovarian stimulation, ovarian tissue cryopreservation may be the only available option for fertility preservation [11,12]. The method, which was first used in 1999 [13], has successfully restored ovarian function, followed by a growing number of live births [14]. This method does not require ovarian stimulation or a sperm donor and may preserve a large number of oocytes. Moreover, recovery of ovarian endocrine functions may be expected, leading to a significant improvement in QoL in young women [15]. The strategy is still considered an experimental method of fertility preservation because of the limited data as compared with oocyte cryopreservation [16]. However, in a recent review, it was reported that frozen-thawed ovarian transplantation has led to approximately 90 live births, with a conception rate of approximately 30% [17]. In a recent prospective observation cohort study, ovarian tissue cryopreservation and transplantation showed comparable efficacy in preserving fertility compared with oocyte vitrification in women undergoing gonadotoxic treatments [18]. Many researchers have suggested that this method should be considered as either an alternative or the primary option for fertility preservation in young women [19].

Ovarian tissue cryopreservation can be performed using slow freezing (also known as equilibrium freezing) and vitrification. In the slow freezing method, ovarian tissue is slowly frozen to approximately −140 °C, followed by the storage of the tissue in liquid nitrogen at −196 °C. No serious tissue deformation has been observed with this method, with the exception of the potential risk for ice crystal formation in the ovarian tissue that may mechanically injure the cells [20]. Vitrification is characterized by the instant solidification of the solution as a result of increased viscosity during cooling with higher concentrations of cryoprotectants than the concentrations used in slow freezing [21]. Advantages of vitrification for oocyte or embryo include a lower risk of ice crystal formation, reduced processing time, and inexpensive equipment. However, very little information is available for ovarian tissue cryopreservation using vitrification [22]. Most reported live births from cryopreserved ovarian tissue have resulted from the use of the slow freezing method [23]. More studies are warranted to investigate the efficiency of the vitrification method [24], given the limited comparison of the effectiveness of vitrification and slow freezing for ovarian tissue cryopreservation [25].

Here, we compared the efficacy and safety of the two methods for ovarian tissue cryopreservation using histopathological and immunohistochemistry (IHC) analyses after human ovarian tissue transplantation in a xenotransplantation model.

## 2. Results

### 2.1. Histologic Evaluation and Primordial Follicular Counts

Xenografted ovarian tissue fragments survived at the grafted site in mice 4 weeks after transplantation. These included the following groups: 111 samples in fresh ovarian tissues (control), 61 thawed after slow freezing (SF), 69 thawed after vitrification (VT), 86 slow freezing-thawing and xenotransplantation (SF-T), and 86 vitrification-thawing and xenotransplantation (VT-T). The gross formation of vessels was identified between the tissue fragments and grafted sites (Figure 1). The numbers of morphologically intact primordial follicles in each experiment group were 13.24 ± 19.65, 7.38 ± 9.64, 3.94 ± 4.45, 2.66 ± 3.62, and 0.28 ± 0.92 per each section in the control, SF group, VT group, SF-T group, and VT-T group, respectively (Figure 2 and Figure 3A). Compared to the control group, the four cryopreserved and transplanted groups showed a statistically significant decrease in the primordial follicle count. The SF group had a significantly higher follicular count than the VT group (*p* = 0.009). The follicular count was also significantly higher in the SF-T group than in the VT-T group (*p* < 0.001). Figure 3A presents representative photographs of hematoxylin and eosin (H&E) stained cryopreserved and xenotransplanted samples.

### 2.2. Follicular Cell Proliferation and Neo-Vascularization

Figure 3B,C display representative images of the cryopreserved and xenografted human ovarian tissues that were immunohistochemically stained for Ki-67 and CD31, respectively. Ten samples from each group were stained for Ki-67 as well as CD31. The mean number of follicles stained with Ki-67 was significantly different among the different groups (*p* < 0.001): 6.90 ± 5.04, 1.92 ± 1.44, 0.50 ± 0.85, 3.90 ± 2.69, and 0.40 ± 0.70 per each section in the control, SF group, VT group, SF-T group, and VT-T group, respectively. The proportion of CD31-positive area was also significantly different among the different groups (*p* < 0.001): 7.35 ± 2.13, 2.47 ± 1.48, 1.82 ± 1.29, 5.24 ± 1.69, and 1.45 ± 0.64 per each section in the control, SF group, VT group, SF-T group, and VT-T group, respectively. Of note, the tissue samples from the SF-T group displayed a significantly higher number of follicles stained with Ki-67 (*p* = 0.048) and higher proportion of CD31-positive area (*p* = 0.004) compared to the samples from the VT-T group.

### 2.3. Gonadotropin Receptors

Figure 4 displays representative images of ovarian tissue samples from the five groups (control, the SF group, VT group, SF-T group, and VT-T group) immunostained for follicle stimulating hormone receptor (FSHR) and luteinizing hormone receptor (LHR). Five samples from each group were stained with FSHR as well as LHR. The FSHR-positive area and LHR-positive area are shown as red and green color, respectively. The proportion of FSHR-positive area in each experiment group was 68.17 ± 15.95, 9.55 ± 6.73, 4.66 ± 3.82, 44.2 ± 7.54, and 24.1 ± 5.42 (*p* < 0.001), and the proportion of LHR-positive area in each experiment group was 63.71 ± 8.45, 29.34 ± 13.49, 38.33 ± 14.42, 56.75 ± 11.51, and 36.85 ± 8.51 (*p* = 0.001), per each section in the control, SF group, VT group, SF-T group, and VT-T group, respectively. The values were lower in the SF and VT groups compared to the control, SF-T, and VT-T groups. Notably, the FSHR-positive area and LHR-positive area were higher in the transplanted tissue group than those in the freezing group. Among the engrafted tissues, the SF-T group displayed a higher proportion of FSHR-positive area compared to the VT-T group (*p* = 0.020).

### 2.4. DNA Damage

The results of terminal deoxynucleotidyl transferase-mediated dUTP nick-end labeling (TUNEL) analysis for DNA fragmentation are presented in Figure 5. TUNEL assays were performed on 20 samples of ovarian tissue from each group and the resulting slides were examined repeatedly and independently by three researchers. The proportion of TUNEL-positive cells producing green fluorescence was significantly different among the groups (*p* < 0.001), with values of 0.51 ± 0.26, 1.24 ± 0.65, 3.04 ± 2.53, 1.86 ± 1.17, and 4.72 ± 3.72 per each section in the control, SF group, VT group, SF-T group, and VT-T group, respectively. DNA fragmentation increased after cryopreservation compared to the control group, and after cryopreservation and transplantation treatments (SF-T and VT-T) compared with cryopreservation alone (SF and VT). Comparison between the results of cryopreservation showed that the VT group presented more DNA damage than the SF group (*p* = 0.043). Comparison between the results after xenotransplantation revealed increased DNA damage in the VT-T group compared to the SF-T group (*p* = 0.033).

### 2.5. Western Blot Analysis

Western blot analysis was performed to identify cell apoptosis using caspase-3 protein as a marker and β-actin as a control. The molecular weight of β-actin is 43 kDa and that of caspase-3 is 35 kDa in whole form, and 19 kDa and 17 kDa in cleaved forms. A thick cleaved caspase-3 band was observed in samples from the VT group, but was absent in samples from the SF and control groups (Figure 6A). The findings indicated that the process of vitrification and thawing may induce necrosis and cause apoptosis in follicular cells. Western blot analysis was also performed to quantify anti-Müllerian hormone (AMH, 60 kDa) production by using ovarian tissues from the five groups (Figure 6B). The expression of AMH was quantified using the ImageJ program and was higher in the control group compared to the other groups (Figure 6C). The expression of AMH was low in transplanted samples from both cryopreservation techniques (SF-T and VT-T), indicative of the decreased production of AMH after xenotransplantation.

### 2.6. Fine Structural Evaluation by Transmission Electron Microscopy (TEM)

TEM morphological evaluation of primordial follicles was performed in three samples from the SF and VT groups. TEM revealed that many of the follicles from the VT group were deformed and destroyed by ice crystal formation after thawing. Follicles from the SF group and the control group were intact and showed no morphological deformation (Figure 7).

## 3. Discussion

Here we showed that the conventional slow freezing method is more suitable than the vitrification method for ovarian tissue cryopreservation and transplantation, as evidenced by the increased follicle preservation, enhanced follicular cell proliferation and angiogenesis, and decreased DNA damage. These results highlight the potential of ovarian tissue cryopreservation and transplantation for fertility preservation in women requiring cancer treatment. The advantages of ovarian tissue transplantation are not only preserving fertility but also restoring endocrine function in young women after cancer treatment [26].

Assessment of morphologically intact primordial follicles is the most widely used index to determine the efficiency of ovarian tissue cryopreservation [27]. In this study, the intact primordial follicle count was remarkably higher after slow freezing and xenotransplantation than after vitrification and xenotransplantation. Consistent with these results, Gandolfi et al. [22] reported that slow freezing was more efficient than vitrification in the preservation of preantral (primordial and growing follicles) follicles, irrespective of the type of the follicle. Oktem et al. found that slow freezing resulted in a higher number of primordial follicles and produced more AMH than vitrification [28]. However, conflicting results have been reported in an experimental study [29] and in a recent systemic review and meta-analysis of 14 experimental studies that compared vitrification with slow freezing for ovarian tissue cryopreservation. In the latter, no significant difference was observed between the two methods in pooled analyses for the follicular density and proportion of intact primordial follicles; vitrification was associated with significantly less follicular DNA damage and better stromal cell preservation [30]. However, the review also highlighted that the vitrification protocols used in the studies were diverse and lacked standardization [30]. The reason for the discrepancies between the results of the present study and these prior studies is uncertain.

Angiogenesis of the ovarian tissue grafts at the grafted site is essential for follicle survival and integrity [31]. Previous studies have found that ischemic injuries that occur during graft revascularization during the first few days of transplantation were more detrimental to the results of cryopreservation and transplantation than cryoinjury [32]. In the present study, the number of DNA fragments was higher and AMH production seemed to be lower in the SF-T and VT-T samples used for xenotransplantation, compared to those in SF and VT samples. Factors affecting the outcome of transplantation, such as revascularization and cell proliferation, in the ovarian tissue grafts should be assessed to investigate the efficiency of the ovarian tissue cryopreservation and transplantation. CD31 and Ki-67 were used as indicators of revascularization and cell proliferation, respectively, in cryopreserved and xenografted ovarian tissues. The numbers of CD31-positive vessels and Ki-67-positive follicles were higher after slow freezing and xenotransplantation than after vitrification and xenotransplantation. Choi et al. [33] described that the reduction in the expression of angiogenetic factors (vascular endothelial growth factor and angiopoietin-2) decreased after vitrification without dimethyl sulfoxide (DMSO) as compared with that observed after vitrification using DMSO, implying that a high concentration of cryoprotectant may induce osmotic stress during cryopreservation. Further studies are warranted to assess this speculation and to define the most efficient protocol for vascularization and cell proliferation of cryopreserved and grafted ovarian tissues.

Transplantation of cryopreserved ovarian tissues restores endocrine function and achieves live births [34]. To date, however, only a few investigations have compared the results between vitrification and slow freezing after the transplantation of cryopreserved and thawed ovarian tissues [35,36,37,38]. Although cryopreservation followed by transplantation is relevant in clinical settings, most studies have assessed the efficiency of ovarian tissue cryopreservation after warming or in vitro culture [36]. In the study by Herraiz et al. [35], a vitrification protocol containing DMSO, ethylene glycol, sucrose, and serum substitute supplement (SSS) at concentrations similar to those used in the present study was deemed the most efficient protocol in vitro and was compared with slow freezing cryopreservation in vivo through the xenotransplantation of human ovarian tissue into ovariectomized mice. In contrast to our results, these authors showed that vitrification preserved a larger population of primordial follicles than slow freezing after transplantation. Notably, slow freezing grafts showed more vascularized area and cell proliferation as compared to vitrification grafts, although no statistical significance was observed [35]. On the other hand, Amorim et al. [36] showed that two vitrification protocols exhibited better preservation of preantral follicles than the conventional slow freezing method after xenotransplantation into ovariectomized mice; one of these protocols utilized 20% DMSO and 20% ethylene glycol, which was similar to the protocol used in the present study, and the other utilized 10% DMSO and 26% ethylene glycol, which resulted in a higher proportion of primordial follicles than slow freezing.

Responsivity to gonadotropins, FSH and LH is an important factor in the regulation of ovarian function and folliculogenesis. A previous study reported that the expression of folliculogenesis-related genes, including that encoding FSHR were lower in vitrified human ovarian tissues compared to the control group, although there was no statistical significance [39]. Presently, immunofluorescence staining revealed that FSHR and LHR were significantly decreased after the freezing process, either slow freezing or vitrification, compared to the control group. Notably, these levels were restored in the xenotransplanted tissues, and slow freezing seemed to be more favorable in the restoration than vitrification. Further studies are needed to address the gonadotropins receptivity of cryopreserved and transplanted ovarian tissue and their association with serum gonadotropins and sex hormone levels.

There are some limitations in our study. First, this study lacks long-term data. All grafted tissues were retrieved from mice and evaluated 4 weeks after xenotransplantation. However, previous studies have shown that revascularization and cell survival following ischemic damage occur after approximately 5 days from xenotransplantation [40]. Hence, 4 weeks may be a sufficient length of time to observe these endpoints in grafted tissues. Second, we failed to obtain any data on hormonal tests, including FSH and LH, although AMH expression was assessed by western blot analysis. Although the endocrine function restoration after ovarian transplantation was shown in the previous study [41], further studies should investigate and compare the effects of different cryopreservation protocols on the outcome of ovarian transplantation in terms of endocrine function restoration. Third, the final endpoint during the evaluation of the efficiency of ovarian tissue cryopreservation should be pregnancy and live birth rate along with the restoration of endocrine function and QoL in cancer survivors. Therefore, it is essential to confirm the experimental findings of this study in clinical investigations. Finally, this study did not assess the changes in the gene expression levels in human ovaries after the freezing and thawing process that had been reported in the previous studies [42].

## 4. Materials and Methods

### 4.1. Study Design

Human ovarian cortex tissues were obtained from 19 patients (15–32 years old) that underwent benign ovarian surgery. All patients provided informed consent, and none had undergone chemo/radiotherapy before surgery. Ovarian tissue samples from each individual donor were equally divided and prepared for slow freezing and vitrification. Ovarian tissues were punctured using a biopsy punch (Kai Industries Co., Ltd., Seki City, Japan) to generate identical biopsies 4 mm in diameter and 1 mm thick. These ovarian tissue samples were cryopreserved and thawed as described below, and then transplanted in ovariectomized severe combined immunodeficient (SCID) mice. Four weeks after transplantation, the grafts were retrieved from sacrificed mice (Figure 8). The ovarian tissue samples were divided into five groups as follows: fresh ovarian tissues (control group), tissues thawed after slow freezing (SF group), tissues thawed after vitrification (VT group), tissues retrieved after slow freezing-thawing and xenotransplantation (SF-T group), and tissues retrieved after vitrification-thawing and xenotransplantation (VT-T group). This study was conducted in accordance with the Declaration of Helsinki following approval of the institutional review board of the Korea University Anam Hospital, Seoul, Korea (IRB No.: ED11138; 1 Dec 2016).

### 4.2. Slow Freezing Protocol

The ovarian tissue fragments were transferred to a basic solution containing 5% SSS (catalog #99193, Irvine Scientific, CA, USA) in M199 culture medium (catalog #M4530; Sigma-Aldrich, St. Louis, MO, USA). The cryoprotectant was added by sequential dilution in three steps. The tissue fragments embedded in 5% SSS-supplemented M199 culture medium were exposed to 7.5% DMSO (catalog #D2650, Sigma-Aldrich) for 5 min at 4 °C, followed by sequential treatment with 10% and 12.5% DMSO for 15 min each. All procedures were performed on ice. The ovarian tissue fragments were transferred to individual cryotubes containing 1 mL of medium. The cryotubes were cooled in a programmable controlled-rate freezing device (Planer PLC, Middlesex, UK) with a slow freeze protocol as described previously [43] that featured cooling from 4 °C to −7.0 °C at a rate of −2.0 °C/min, followed by manual seeding and cooling to −40.0 °C at a rate of −0.3 °C/min and −140 °C at a rate of −10 °C/min. Finally, the samples were stored at −196 °C in liquid nitrogen.

### 4.3. Vitrification Protocol

Vitrification was conducted as previously described [44]. Ovarian fragments were first incubated in an equilibration solution (ES) consisting of 65 mL HEPES (catalog #15630-080; Gibco, Taiwan), 20 mL SSS, 7.5 mL ethylene glycol, and 7.5 mL DMSO (total concentrations: 7.5% ethylene glycol, 7.5% DMSO, and 20% SSS) for 25 min at room temperature (24 ± 2 °C). After 25 min, the ES was removed with a sterilized gauze and each sample was transferred to a conical tube containing vitrification solution (VS) comprised of 20% ethylene glycol, 20% DMSO, 0.5 M sucrose (catalog #S1888, Sigma-Aldrich), and 20% of SSS in HEPES buffer and incubated for 15 min at room temperature. After incubation in VS, the tissues were placed on a sterilized gauze to remove excess VS. The tissues were separately placed in vials and quickly plunged into fresh liquid nitrogen. The samples were checked for translucency. Samples that were not translucent were considered to have ice crystals and were therefore excluded. The vials of the selected vitrified tissues were sealed and stored in a liquid nitrogen storage tank.

### 4.4. Slow Freezing-Thawing Protocol

Tissue thawing was carried out as described previously [43]. The stored cryotube vials were transferred from liquid nitrogen to a shaking water bath at 37 °C. Upon complete thawing of the vials, half of the supernatant was removed and replaced with the same volume of washing solution containing 5% DMSO in basic medium. The vial was incubated for 10 min at room temperature. After incubation, half of the supernatant was removed again and replaced with the same volume of washing solution. The vial was incubated for 5 min at room temperature.

### 4.5. Vitrification-Thawing Protocol

The vitrification freezing vial was moved from the storage tank, and tissues were placed into a thawing solution (55 mL HEPES, 20 mL SSS, and 34.24 g sucrose). The tissues were allowed to completely thaw and were transferred to diluent solution (65 mL HEPES, 20 mL SSS, and 17.12 g sucrose) for 5 min at room temperature. Following incubation, tissues were transferred into a washing solution (40 mL HEPES and 10 mL SSS) for 5 min at room temperature. The washing step was repeated.

### 4.6. Xenotransplantation into SCID Mice

Thawed human ovarian tissues were washed twice with phosphate buffered saline (PBS) and stored in normal saline. Sixty female C.B-17/Fox Chase SCID mice, 6 weeks of age, and 52 female SCID mice were ovariectomized before xenotransplantation under anesthesia via a small incision in the body wall that was sutured with 6-0 silk thread. The human frozen/thawed ovarian tissues were transplanted onto the back muscle of the mice. This study followed the animal welfare guidelines approved by the Institutional Animal Care and Use Committee of the Korea University Anam Hospital.

### 4.7. Histologic Evaluation

Four weeks after transplantation, the grafts were retrieved from sacrificed mice. Twenty ovarian tissue samples in each of the five groups (control, SF, VT, SF-T, and VT-T groups) were fixed with 4% formaldehyde and subjected to histological evaluation. After fixation, tissues were washed three times with PBS and dehydrated stepwise in graded ethanol solutions (50%, 70%, 80%, 90%, and 100%). After routine paraffin embedding, samples were serially sectioned (2 μm thick) and stained with H&E, followed by microscopic examination to determine follicle count and density. Three researchers independently examined the samples repeatedly. To avoid repetition, only follicles containing an oocyte with a visible nucleus were counted. The follicles were classified according to their developmental stages based on the granulosa cell morphology. A primordial follicle was defined as an oocyte surrounded by a single fusiform granule cell, while a primary follicle was an oocyte surrounded by a single layer of cuboidal granulosa cells. Secondary follicles were surrounded by six to eight layers of cuboidal granulosa cells with no visible antrum [45]. Morphologically intact follicles were identified based on the oocyte integrity by an experienced pathologist.

### 4.8. Immunohistochemistry Evaluation

IHC staining with Ki-67 and CD31 was performed to evaluate cell proliferation and intensity of angiogenesis in cryopreserved and xenografted human ovarian tissues. Twenty samples in each of the five groups were evaluated with Ki-67 staining, and the same number of samples were evaluated with CD31 staining. For IHC staining, unstained and paraffin-embedded ovarian tissue slides were deparaffinized with xylene and rehydrated with a series of ethanol solutions (50%, 70%, 80%, 90%, and 100%). The slides were left in a preheated target retrieval solution (Tris-ethylenediaminetetraacetic acid [EDTA], pH 8.0; Dako, USA) for 13 min in a microwave for heat-induced epitope antigenic retrieval. The samples were blocked with peroxide blocking solution for 8 min and washed with the washing buffer (1× Tris–Tween-20 buffer in deionized water). After blocking, sections were incubated with antibody to Ki-67 (1:100 dilution, Cell Marque, 275R-14) or CD31 (1:200 dilution, Abcam, UK) for 1 h at room temperature. Slides were incubated for 20 min with GBI Polink-2 and rabbit antibodies. After washing, the sections were treated with EnVision+ horseradish peroxidase (HRP) and liquid 3,3′-diaminobenzidine tetrahydrochloride (DAB)+ substrate and counterstained with Mayer’s hematoxylin (Scytek, Logan, UT, USA). The slides were dehydrated and mounted before their microscopic examination. Using the microscopic examination, the number of follicles stained with Ki-67 was counted, while the proportion of area stained with CD31 was quantified using the ImageJ program.

### 4.9. Immunofluorescence Staining Evaluation

Immunofluorescence staining of the ovarian tissues for FSHR and LHR was also performed. Each tissue was fixed with 4% paraformaldehyde (Sigma-Aldrich) and embedded using paraffin. Paraffin block was sectioned at 5 µm thickness using a microtome (Leica, Allendale, NJ, USA) and the sections were placed on glass slides. The sections were deparaffinized by xylene (Sigma-Aldrich) and rehydrated by incubation in a series of ethanol solutions. Dehydrated tissues were blocked using 3% goat serum (Vector Laboratory) for 12 h at 4 °C and primary antibodies, rabbit anti-human FSHR (1:100, Abcam), and rabbit anti-human LHR (1:100, Abcam) were incubated for 12 h at 4 °C. After washing two times with PBS and 0.3% Triton X100 (PBST), the sections were treated with secondary antibodies (goat IgG anti-rabbit-488 and -594, 1:100, Molecular Probes, Eugene, OR, USA) for 2 h at room temperature. Following washing three times with PBST, the slides were mounted with mounting medium (Vector Laboratories, USA). The fluorescence images were captured using an EVOS FL microscope (Thermo Fisher Scientific, Waltham, MA, USA). The proportion of stained area was quantified using the ImageJ program (NIH, Bethesda, MD, USA).

### 4.10. TUNEL Assay

DNA fragmentation was analyzed using the TUNEL assay. After deparaffinization, the samples were rehydrated with graded ethanol and immersed in 4% formaldehyde in PBS (catalog #AB216603; Hyclone, USA) for 15 min. The samples were processed using the DeadEnd™ Fluorometric TUNEL System (catalog #G3250, Promega, USA). The nuclei were stained with 1 mg/mL 4′,6-diamidino-2-phenylindole (DAPI) and mounted with VECTASHIELD mounting medium (catalog #H-1400; Vector Laboratories). TUNEL-positive cells produced green fluorescence and immunofluorescence images were obtained using a fluorescence microscope (Olympus, Japan) at 400× magnification. The proportion of TUNEL-positive area was quantified using the ImageJ program.

### 4.11. Western Blot

Western blot analyses were performed to evaluate cell apoptosis based on caspase-3 expression and to analyze AMH production by human ovarian tissues. Tissue lysates were obtained using 1× radioimmunoprecipitation assay (RIPA) buffer (catalog #BRI-9001; BioPrince, South Korea) supplemented with protease inhibitor (catalog #04693159001; Roche, Switzerland) and phosphatase inhibitor (catalog #04906837001; Roche). The extracted protein concentration was measured according to the well-established Bradford protocol. Equal amounts of total protein (30 μg) were boiled in 1× loading dye for 10 min and subjected to 15% sodium dodecyl sulfate-polyacrylamide gel electrophoresis (SDS-PAGE). Protein bands were transferred to polyvinylidene fluoride (PVDF) membranes (Bio-Rad Laboratories, USA) and the membranes were blocked with 1× TBS (catalog #BTT-9120; Tech & Innovation, South Korea) containing 5% skim milk (catalog #7B48028, Bioshop Canada, Ontario) and 0.05% Tween-20 at room temperature for 3 h. The membranes were probed with anti-caspase-3 (1:100 dilution, catalog #9662; Cell Signaling Technology, USA), anti-AMH (1:100 dilution, catalog #ab103233; Abcam), and anti-β actin (1:1000, catalog #sc-47778; Santa Cruz Biotechnology, USA) primary antibodies at 4 °C overnight with gentle shaking. Following incubation, the membranes were washed three times with 1× TBS containing 0.05% Tween-20 and incubated with goat anti-rabbit secondary antibody (1:5000 dilution, catalog #ab6721; Abcam) in 1× TBS containing 3% skim milk and 0.05% Tween-20 at room temperature for 60 min. Immunoreactive proteins were visualized by chemiluminescence using Clarity Western ECL substrate (catalog #1705060, Bio-Rad Laboratories) and detected on medical X-ray film blue (Agfa-Gevaert, Belgium). Western blot was repeated five times, both for caspase-3 and AMH analyses. The expression of protein was quantified using the ImageJ program.

### 4.12. TEM

Tissues were fixed with 2.5% glutaraldehyde in 0.1 M phosphate buffer at 4 °C overnight. Samples were then washed and fixed with 1% osmium tetroxide, followed by dehydration and embedment in Epon mixture (Polybed 812 embedding kit/DMP-30; catalog #08792-1, Polysciences, Inc., USA). The embedded samples were incubated at 65 °C in a dry oven for 48 h to polymerize the resin. Sections (1 μm thickness) were obtained using a Reichert-Jung Ultracut E ultramicrotome (Leica, Japan) and stained with toluidine blue. Sections 60 nm in thickness were collected on 200 mesh cooper grids. These were stained with uranyl acetate/lead citrate and their morphologies were analyzed using a model H-7500 apparatus (Hitachi, Japan) at an operating voltage of 80 kV.

### 4.13. Statistical Analyses

The results of follicle counts, number of Ki-67 positive cells, proportion of CD31 positive area, and TUNEL-positive cells within the samples were compared with Student’s independent *t*-test or analysis of variance (ANOVA) using SPSS version 12.0 software (SPSS Inc., USA). A *p*-value < 0.05 was considered statistically significant.

## 5. Conclusions

Ovarian tissue cryopreservation and transplantation is an efficient and safe method for fertility preservation and endocrine function recovery in cancer survivors of reproductive age. Although vitrification has shown promising results in recent experimental studies, the present study using a xenotransplantation model indicates that the slow freezing method is superior to vitrification in terms of primordial follicle preservation, vascularization, follicular cell proliferation, DNA damage, and AMH expression. Slow freezing should be still regarded as an effective option for ovarian tissue preservation for women with cancer who are interested in preserving their fertility, and further studies should be directed towards standardization of the vitrification method.

## Figures and Tables

**Figure 1 ijms-20-03346-f001:**
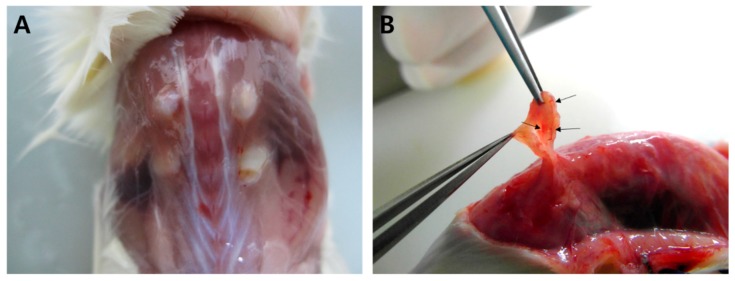
Data on grafted tissue. (**A**) Gross finding of grafted human ovarian tissue at 4 weeks after xenotransplantation on the back muscle of ovariectomized SCID mice. (**B**) The grafted tissue survived and the formation of vessels between the tissue fragment and grafted site was identified (arrows).

**Figure 2 ijms-20-03346-f002:**
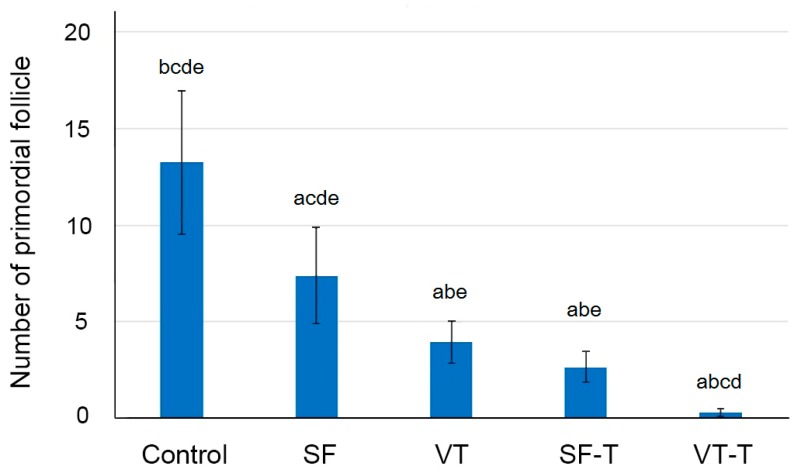
The mean number of primordial follicles in the five study groups with 95% confidence intervals. Student’s independent *t*-test showed that the mean number of follicular counts was significantly different among the different groups (letters above bars indicate the another groups statistically different from the one, *p* < 0.05): a, different from the control group; b, different from the SF group; c, different from the VT group; d, different from the SF-T group; e, different from the VT-T group. Abbreviations: SF, cryopreserved with slow freezing technique and thawed group; VT, cryopreserved with vitrification technique and thawed group; SF-T, the group xenotransplanted after slow freezing and thawing; VT-T, the group xenotransplanted after vitrification and thawing.

**Figure 3 ijms-20-03346-f003:**
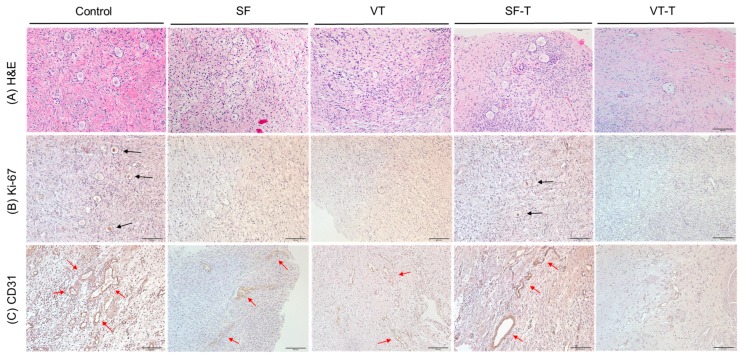
Histopathological findings of the human ovarian tissues after cryopreservation and xenotransplantation. (**A**) Hematoxylin and eosin staining (H&E) of human ovarian tissues in the five groups. (**B**) Histological features of human ovarian tissues in the five groups evaluated with immunohistochemistry staining for Ki-67, a marker of cell proliferation. Black arrows indicate the cells stained with Ki-67. (**C**) Histological features of human ovarian tissues in the five groups evaluated with immunohistochemistry staining for CD31, a marker of angiogenesis. Red arrows indicate the cells stained with CD31. Original magnification ×200. Abbreviations: SF, cryopreserved with slow freezing technique and thawed group; VT, cryopreserved with vitrification technique and thawed group; SF-T, the group xenotransplanted after slow freezing and thawing; VT-T, the group xenotransplanted after vitrification and thawing.

**Figure 4 ijms-20-03346-f004:**
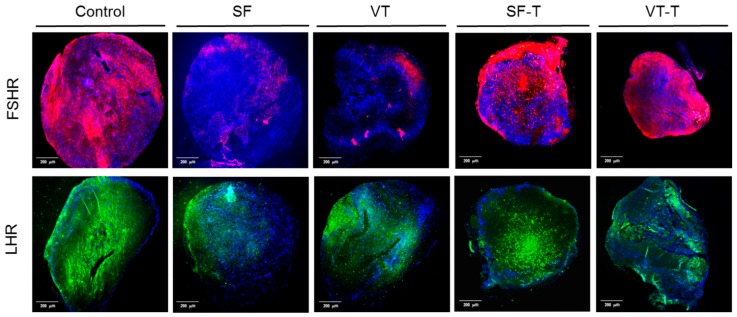
Immunofluorescence staining of ovarian tissues with follicle stimulating hormone receptor (FSHR) and luteinizing hormone receptor (LHR). Positive area for those receptors are shown as red and green, respectively. The FSHR-positive area and LHR-positive area were higher in the transplanted tissue group (SF-T and VT-T) than those in the freezing group (SF and VT). Among the engrafted tissues, the SF-T group displayed a higher proportion of FSHR-positive area compared to the VT-T group. Abbreviations: FSHR, follicle stimulating hormone receptor; LHR, luteinizing hormone receptor; SF, cryopreserved with slow freezing technique and thawed group; VT, cryopreserved with vitrification technique and thawed group; SF-T, the group xenotransplanted after slow freezing and thawing; VT-T, the group xenotransplanted after vitrification and thawing.

**Figure 5 ijms-20-03346-f005:**
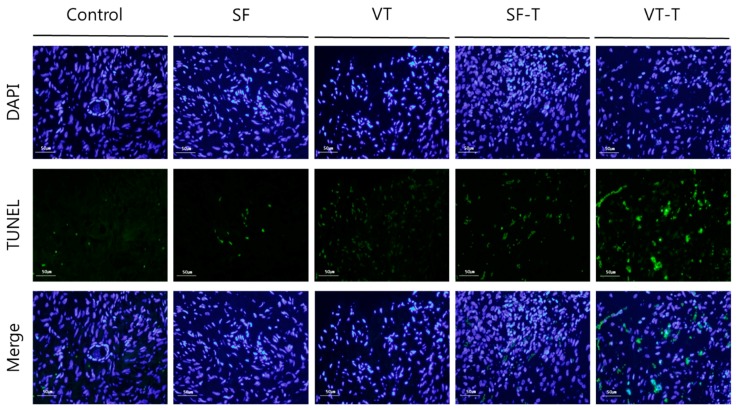
TUNEL assay results for DNA double-strand breaks in human ovarian tissues. Green fluorescence in the images indicates DNA fragmentation in the ovarian tissues. The proportion of TUNEL-positive area corresponding to DNA damage appeared to be higher in the SF and VT groups compared to those in the control group, in the SF-T and VT-T groups compared those in SF and VT groups, and in the VT-T group compared those in the SF-T group. Abbreviations: SF, cryopreserved with slow freezing technique and thawed group; VT, cryopreserved with vitrification technique and thawed group; SF-T, the group xenotransplanted after slow freezing and thawing; VT-T, the group xenotransplanted after vitrification and thawing; TUNEL, terminal deoxynucleotidyl transferase-mediated dUTP nick-end labeling; DAPI, 4′,6-diamidino-2-phenylindole.

**Figure 6 ijms-20-03346-f006:**
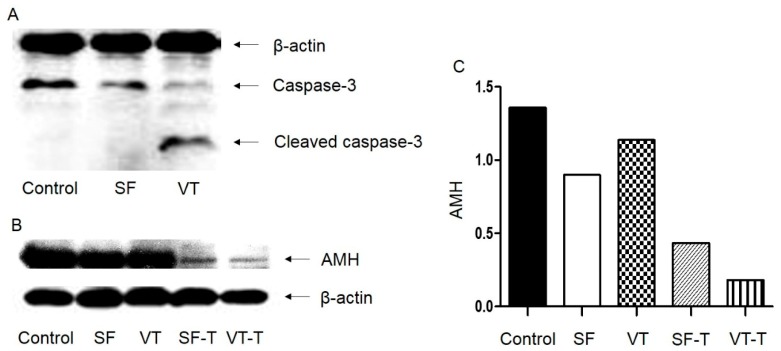
Western blot analysis to evaluate cell apoptosis based on caspase-3 expression (**A**) and anti-Müllerian hormone production by human ovarian tissues (**B**,**C**). Molecular weight: β-actin, 43 kDa; caspase-3, whole form, 35 kDa; caspase-3, cleaved forms, 19 kDa and 17 kDa; AMH, 60 kDa. Abbreviations: SF, cryopreserved with slow freezing technique and thawed group; VT, cryopreserved with vitrification technique and thawed group; SF-T, the group xenotransplanted after slow freezing and thawing; VT-T, the group xenotransplanted after vitrification and thawing; AMH, anti-Müllerian hormone.

**Figure 7 ijms-20-03346-f007:**
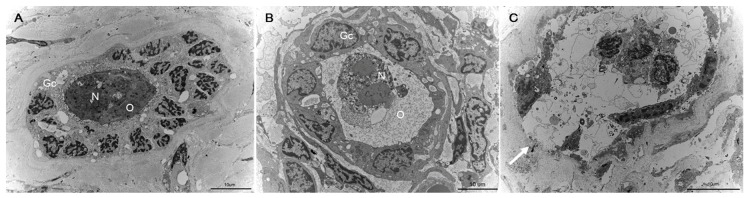
Morphological analysis of primordial follicles from human ovarian tissues using transmission electron microscopy. (**A**) The primordial follicle from the fresh ovarian tissue (control group) (**B**) The primordial follicle from cryopreserved with slow freezing and thawed group. A round oocyte (O) tightly connected to the surrounding granulosa cells (Gc) is evident. The cytoplasmic membrane and envelope of nucleus (N) are intact. (**C**) Follicle from vitrified and thawed group. The follicle from vitrification group was deformed and destructed by ice crystal formation after thawing (white arrow), whereas the follicle from the control group and the slow freezing group were intact.

**Figure 8 ijms-20-03346-f008:**
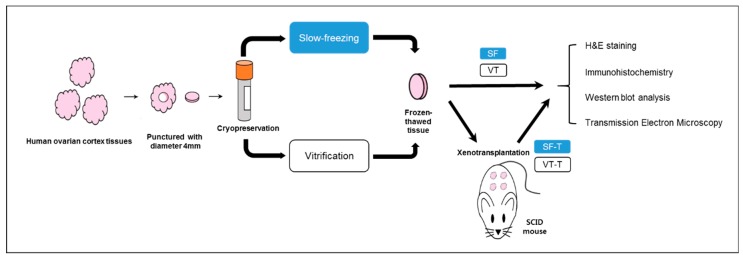
Schematic representation showing the experimental design of this study. Human ovarian cortex tissues were punctured to make same size with diameter 4 mm, and those samples were cryopreserved with slow freezing or vitrification methods. After 4 weeks, the thawed samples underwent histological analyses or were transplanted into SCID mice. The transplanted samples were retrieved after 4 weeks and underwent histological analyses. Abbreviations are: SF, cryopreserved with slow freezing technique and thawed group. VT, cryopreserved with vitrification technique and thawed group. SF-T, the group xenotransplanted after slow freezing and thawing. VT-T, the group xenotransplanted after vitrification and thawing, and SCID, severe combined immunodeficient.

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
