# Peer review of "Comparison between Slow Freezing and Vitrification for Human Ovarian Tissue Cryopreservation and Xenotransplantation"

_ijms, 2019, doi:10.3390/ijms20133346_

Reviewer 1 Report

Dear authors,

I have read your manuscript "Comparison Between Slow Freezing and Vitrification for Human Ovarian Tissue Cryopreservation and Xenotransplantation" with great interest.

I have found this research very remarkable, in my opinion having the high potential for the future in the field of Reproductive biology research and clinical practice.

However, I have also detected some discrepancies/errors in your manuscript. In my opinion many things have to be corrected or added. I am going to explain my opinions and suggestions/recommendations in common order.

Abstract 

lns 18 - 19 - please add statistical significance

lns 20 - 21 - comment as above

ln 25 - 26 - why authors mention only part of their results?

Introduction 

This part is generally well written. I do not have any comments.

Results 

2.1.

The whole part should be in section Material and Methods. Moreover, the creation of the graphic experimental design from this part will be really beneficial.

2.2.

in figure 1 part B I suggest to place a more detailed picture and also to change the color of arrows in order to be more obvious and more "catchy".

in figure 2 statistical significances are completely missing. In related section authors describing significant differences, however, in the figure, they are not displayed. 

2.3.

lns 113 - 114 - please state the number of counted cells. In my opinion, only write ...a higher proportion of positive cells is vague

Fig. 3 - I suggest adding arrows or other characters to pictures to depict them better 

2.4.

lns 126 - 127 How many samples authors analyzed? Are they in dublets? How many cells were counted? This data are very important.

Fig. 4 - If control in the first column means only samples of fresh ovaries. The negative control is missing!! Also, positive control is missing, nevertheless, it is usually optional.

I also recommend better description of figure relating what we can see on the picture and also to color fluorescence and what it means. Figures have to be described in a way that is sufficient to make them clear also when they stand alone (without text).

What was magnification?

2.5.

In this section, I strongly suggest to authors state molecular weights.

ln 144 - What does mean slightly higher? Do the authors performed any quantification? 

ln 145 - From which authors infer that these bands in rows SF-T and VT-T are real bands and not non-specific reaction or background?

ln 146 - check the abbreviation - AMT

Fig. 5

I strongly recommend to the authors to add molecular weights where protein bands were transferred.

Why actin bands are in the upper part of section B?. For me, it is not logical to have bands with lower weight upper than those with higher molecular mass.

In rows of SF-T, VT-T there are obvious bands. Are they also in control, SF and VT?

The negative control is missing!!

2.6.

Same question as above, how many samples/follicles were counted?

Fig. 6.

Again, control is missing. Why the authors did not place picture of freshly obtained ovaries?

I recommend to authors better description of pictures.

Discussion

ln 173 - please write correctly the reference

lns 180 - 190 - In my opinion, this statement does not have  in absence of controls in TUNEL and proteomic experiments

lns   188 - 192 - I am lacking the logical continuity of sentences in this section.

ln 207 - please correct the reference

ln 215 - please correct the reference

ln 218 - these concentrations are the same as authors used in their study? What were concentrations of the second protocol for vitirification?

ln 227 - Do authors mean AMH secretion? Without negative control and hormonal testing of animals, I would not be sure with this statement.

Material and Methods

4.2.

ln 250 - In which steps modifications are? 

ln 261 - spell check  the word sealing

4.6.

How many samples were analyzed? Are they counted in replicates?

4.7.

ln 298 - In my opinion, it is better to not use abbreviations in the heading of the section.

ln 302 - same series of the ethanol solutions as in section 4.6. ln 288 were used?

how many samples/cells were analyzed?

4.8.

Did authors prepare also negative and positive controls? This is essential for interpreting results in a good way.

ln 319 - 320 - Which cells were counted? 

In my opinion, this part is not clear. In the section Results, authors described generally the positive signal from cells in the tissue. 

I suggest correcting parts of section in Material and Methods and Results relating this topic.

ln 321 - What was the magnification?

4.9.

Did authors prepared negative controls?

How many replicates were done?

How was the analysis of signal was performed?

4.11.

353 - Only follicle count was statistically analyzed? 

Conclusion 

360 - 362 - Why authors only mention part of their results, when they e.g. in lns 189 -191 presenting differences in DNA frag and in ln 227 stating that they observed endocrine function restoration.

362- I disagree with this statement in that form. In my opinion, it is presumptuous too much.

Author Response

Thank you very much for your March 14 2019 decision letter. We appreciate the opportunity to revise and resubmit our manuscript. We have carefully read the reviewer’s constructive comments and have responded to them point-by-point below. And the revisions are highlighted using the "Track Changes" function in our revised manuscript file. We hope that our responses to the reviewer’s comments are adequate.

Thank you for your comments. We have taken into consideration your comments and revised the manuscript as follows.

Comments:

Dear authors,

I have read your manuscript "Comparison Between Slow Freezing and Vitrification for Human Ovarian Tissue Cryopreservation and Xenotransplantation" with great interest.

I have found this research very remarkable, in my opinion having the high potential for the future in the field of Reproductive biology research and clinical practice.

However, I have also detected some discrepancies/errors in your manuscript. In my opinion many things have to be corrected or added. I am going to explain my opinions and suggestions/recommendations in common order.

Abstract

lns 18 - 19 - please add statistical significance

Answer: We thank the reviewer for pointing this out and we have added the statistical significance after that statement in the abstract of the revised manuscript. (Page 1, line 19)

lns 20 - 21 - comment as above

Answer: We thank the reviewer for pointing this out and we have added the statistical significance after that statement in the abstract of the revised manuscript. (Page 1, line 21-22)

ln 25 - 26 - why authors mention only part of their results?

Answer: We thank the reviewer for pointing this out. To summarize our findings more comprehensively, we have change that sentence as following: “Therefore, our findings indicate that slow freezing for ovarian tissue cryopreservation is superior to vitrification in terms of follicle survival and growth after xenotransplantation.” (Page 1, line 26-28)

Introduction

This part is generally well written. I do not have any comments.

Results

2.1.

The whole part should be in section Material and Methods. Moreover, the creation of the graphic experimental design from this part will be really beneficial.

Answer: We thank the reviewer for pointing this out and we agree with the reviewer. Therefore, this part has moved completely into 4.1. Study design part of the Materials and Methods section in the revised manuscript (Page 9, line 278-292). And according to the reviewer’s suggestion, we have added the Figure 7 which presented the whole process of our experiment in the revised manuscript (Page 9).

2.2.

in figure 1 part B I suggest to place a more detailed picture and also to change the color of arrows in order to be more obvious and more "catchy".

Answer: We appreciate for reviewer’s kind suggestion. We have updated the picture with better resolution in Figure 1B and changed the color of arrows in it (Page 3).

in figure 2 statistical significances are completely missing. In related section authors describing significant differences, however, in the figure, they are not displayed.

Answer: We thank the reviewer for pointing this out and we agree with the reviewer. Therefore, we have indicated the statistical significance using the letters above the bars in the graph and explained about it in the figure caption of the revised manuscript (Page 3, line 106-114).

2.3.

lns 113 - 114 - please state the number of counted cells. In my opinion, only write ...a higher proportion of positive cells is vague

Answer: We thank the reviewer for pointing this out and we agree with the reviewer. Therefore, we have added more detailed explanation of our experiment and the quantified results with statistical significance in the revised manuscript, as following: “Ten samples from both the SF-T and VT-T groups were stained with Ki-67, as well as CD31. The number of follicles stained with Ki-67 was significantly higher in the slow freezing groups compared to that from the vitrification groups (4.0 ± 2.76 versus 1.0 ± 1.16, P=0.006). Moreover, the tissue samples from the slow freezing groups displayed a higher proportion of CD31-positive area than samples from the vitrification groups (1.11 ± 0.40 versus 0.29 ± 0.13, P<0.001).” (Page 4, line 120-125) Because only a few follicles in VT-T samples were preserved and stained, the differences of degree of immunostaining between the SF-T and VT-T samples are grossly identified in the representative images in Figure 3. To view more intuitively the comparative results of immunostaining with Ki-67 and CD31, we have added the arrows pointing the stained cells in Figure 3 of the revised manuscript (Page 4).

Fig. 3 - I suggest adding arrows or other characters to pictures to depict them better

Answer: We thank the reviewer for pointing this out and we have added the arrows pointing the stained cells in Figure 3 of the revised manuscript.

2.4.

lns 126 - 127 How many samples authors analyzed? Are they in dublets? How many cells were counted? This data are very important.

Answers: Thank you for your comment and suggestions. Twenty samples of ovarian tissue in each of the five groups underwent TUNEL assay and those slides were examined repeatedly by three researchers. We performed TUNEL assay to compare the proportion of DNA fragmentation among the treated (freezing with or without transplantation) samples and the non-treated samples, and the difference was observed in the results (images) presented in Figure 4. We have indicated the number of samples we examined with TUNEL assay in the Result section of the revised manuscript, as following: “TUNEL assays were performed on 20 samples of ovarian tissue from each group and the resulting slides were examined repeatedly by three independent researchers.” (Page 5, line 140-142)

Fig. 4 - If control in the first column means only samples of fresh ovaries. The negative control is missing!! Also, positive control is missing, nevertheless, it is usually optional.

Answer: Thank you for your comment and suggestions. “Negative control” is a treatment that by definition is expected not to have any effect. “Positive control” is treatment with a well-known chemical that is known to produce the expected effect with the assay that researchers are studying. The American Society of Clinical Oncology recognizes fertility preservation as a key survivorship issue and utilizes fertility treatment as a quality care measure. Currently, the primary fertility preservation treatment options include embryo cryopreservation, oocyte cryopreservation and ovarian tissue cryopreservation (OTC). Clinically, OTC and transplantation is still considered as an experimental method. For prepubertal girls, however, OTC is the only option available since their ovaries will not respond to controlled ovarian hyperstimulation necessary for oocyte retrieval. It is also the only option for women who cannot delay starting chemotherapy.

In this study authors would like to compare the outcomes of two cryopreservation methods (slow vs. vitrification) and xenotransplantation. Thus, we would like to consider the fresh ovarian tissue as a negative control in this study and currently, positive control is not available since human OTC method is not established at the present time. Authors appreciate for your comments and suggestions.

I also recommend better description of figure relating what we can see on the picture and also to color fluorescence and what it means. Figures have to be described in a way that is sufficient to make them clear also when they stand alone (without text).

Answer: We thank the reviewer for pointing this out and we agree with the reviewer. Therefore, we have modified the details of Figure 4 and we have presented more explanation about the result of TUNEL assay in the figure caption, as following: “Green fluorescence in the images indicates DNA fragmentation in the ovarian tissues. The proportion of TUNEL-positive area corresponding to DNA damage appeared to be higher in the SF and VT groups compared to those in control group, in the SF-T and VT-T groups compared those in SF and VT groups, and in the VT-T group compared those in SF-T group.” (Page 5, line 153-157)

What was magnification?

Answer: The magnification was 400x and we have added the scale bars in the pictures of Figure 4. (Page 5)

2.5.

In this section, I strongly suggest to authors state molecular weights.

Answer: We thank the reviewer for pointing this out and we have indicated the molecular weights of the proteins in the revised manuscript: beta-actin, 43kDa; caspase-3, whole form, 35kDa; caspase-3, cleaved forms, 19 kDa and 17 kDa; AMH, 60 kDa. (Page 6, line 164-165, 168)

ln 144 - What does mean slightly higher? Do the authors performed any quantification?

Answer: The expression of AMH was quantified using the ImageJ program and we have newly presented it as a graph in Figure 5C. We have also modified the statement explaining the results of AMH expression as following: “The expression of AMH was quantified using the Image-J program and was higher in the control group compared to the other groups (Fig. 5C). The expression of AMH was low in transplanted samples from both cryopreservation techniques (SF-T and VT-T), indicative of the decreased production of AMH after xenotransplantation.” (Page 6, line 169-173) On the other hand, we did not present the quantitative graph of Caspase-3 expression because the band of cleaved caspase-3 was only shown in VT group.

ln 145 - From which authors infer that these bands in rows SF-T and VT-T are real bands and not non-specific reaction or background?

Answer: We repeated the Western blot analysis three times and could observe the similar results among those test. To present our results more clearly, we have changed the picture of AMH expression to another one with better visibility (Fig. 3B) and we have added the graph showing the quantitative comparison of AMH expression between the groups (Fig 3C).

ln 146 - check the abbreviation - AMT

Answer: We have corrected it into AMH. (Page 6, line 173)

Fig. 5

I strongly recommend to the authors to add molecular weights where protein bands were transferred.

Answers: Thank you for your kind suggestions. We have added the statements of molecular weights in the caption of Figure 5. (Page 6, line 177-178)

Why actin bands are in the upper part of section B?. For me, it is not logical to have bands with lower weight upper than those with higher molecular mass.

Answer: We thank the reviewer for pointing this out and we agree with the reviewer. Therefore, we have shifted the location of actin bands and AMH bands in Figure 5B.

In rows of SF-T, VT-T there are obvious bands. Are they also in control, SF and VT?

Answer: We have altered this picture to more clear one (Fig. 3B).

The negative control is missing!!

Answer: (We have answered about the negative control and positive control in our study at the above answer to the comment for Fig. 4)

2.6.

Same question as above, how many samples/follicles were counted?

Answers: We thank the reviewer for pointing this. Three samples of ovarian tissue in each of two groups (SF and VT groups) underwent TEM examination and those were examined repeatedly by three researchers. We have explained it in the revised manuscript as following: “TEM morphological evaluation of primordial follicles was performed in three samples from each group: SF and VT group.” (Page 6, line 184)

Fig. 6.

Again, control is missing. Why the authors did not place picture of freshly obtained ovaries?

Answer: Thank you for your comment and suggestions. TEM morphological evaluation of primordial follicles was performed in three samples from each group: SF and VT group. And we have found that the primordial follicles in VT group were deformed and destructed as we presented in the Figure 6, whereas the primordial follicles in SF groups were morphologically intact. As the reviewer pointed out, the control is missing in this images. However, we performed TEM examinations to compare the morphological findings of follicles between SF and VT group, and the differences between them might be clearly identified in the Figure 6.

I recommend to authors better description of pictures.

Answer: Thank you for your kind suggestions. We have added the marks to indicate the structure of primordial follicles in the TEM pictures of Figure 6, and we also added more detailed description of pictures in the figure caption. (Page 7, line 190-196)

Discussion

ln 173 - please write correctly the reference

Answer: We have changed that sentence as following: “Gandolfi et al. [22] reported that slow freezing was more efficient than vitrification in the preservation of preantral (primordial and growing follicles) follicles, irrespective of the type of the follicle.” (Page 7, line 206-208)

lns 180 - 190 - In my opinion, this statement does not have  in absence of controls in TUNEL and proteomic experiments

Answer: We thank the reviewer for pointing this out and we agree with the reviewer. Therefore, we have changed that statement as following: “In the present study, the number of DNA fragments was higher and AMH production seemed to be lower in SF-T and VT-T samples, on which the xenotransplantation was conducted, compared to those in SF and VT samples.” Page 8, line 225-227)

lns   188 - 192 - I am lacking the logical continuity of sentences in this section.

Answer: We thank the reviewer for pointing this out and we agree with the reviewer. Therefore, we have changed that statement as following: “Factors affecting the outcome of transplantation such as revascularization and cell proliferation in the ovarian tissue grafts should be assessed to investigate the efficiency of the ovarian tissue cryopreservation and transplantation.” (Page 8, line 227-230)

ln 207 - please correct the reference

Answer: We thank the reviewer for pointing this out and we have corrected it. (Page 8, line 245)

ln 215 - please correct the reference

Answer: We thank the reviewer for pointing this out and we have corrected it. (Page 8, line 253)

ln 218 - these concentrations are the same as authors used in their study? What were concentrations of the second protocol for vitirification?

Answer: To describe the protocols and results of that study more clearly, we have changed that statement as following: “On the other hand, Amorim et al. [35] showed that two vitrification protocols exhibited better preservation of preantral follicles than the conventional slow freezing method after xenotransplantation into ovariectomized mice; one of these protocols utilized 20% DMSO and 20% ethylene glycol, which was similar to the protocol used in our present study, and the other utilized 10% DMSO and 26% ethylene glycol, which resulted in a higher proportion of primordial follicles than slow freezing.” (Page 8, line 253-262)

ln 227 - Do authors mean AMH secretion? Without negative control and hormonal testing of animals, I would not be sure with this statement.

Answer: We thank the reviewer for pointing this out and we agree with the reviewer. Therefore, we have changed that statement as following: “Although the endocrine function restoration after ovarian transplantation was shown in the previous study [39], further studies should investigate and compare the effects of different cryopreservation protocols on the outcome of ovarian transplantation in terms of endocrine function restoration.” (Page 9, line 269-272)

Material and Methods

4.2.

ln 250 - In which steps modifications are?

Answer: We thank the reviewer for pointing this out. We have reviewed this part and have found out that ‘with minor modifications’ was miswriting. The vitrification protocol in our study followed the protocol described previously by Kagawa et al., and there was no differences in the concentration of solutions and order of processes of experiment between that study and ours. Therefore, we have modified that statement as following: “The vitrification was conducted as previously described by Kagawa et al. [41]. (Page 10, line 319-320)

ln 261 - spell check  the word sealing

Answer: We thank the reviewer for pointing this out. It has been corrected into ‘sealed’ in the revised manuscript. (Page 10, line 330)

4.6.

How many samples were analyzed? Are they counted in replicates?

Answer: Twenty samples in each group of ovarian tissue underwent histological evaluation. Three researches examined those samples repeatedly. We have described it in the ‘4.7. Histologic evaluation’ part of the revised manuscript. (Page 10, line 354-355)

4.7.

ln 298 - In my opinion, it is better to not use abbreviations in the heading of the section.

Answer: We thank the reviewer for pointing this out and we have changed it into ‘immunohistochemistry evaluation’. (Page 11, line 368)

ln 302 - same series of the ethanol solutions as in section 4.6. ln 288 were used?

Answer: We used the same series of the ethanol solutions (50%, 70%, 80%, 90%, and 100%) in both examinations, and we have indicated this information in the ‘4.8. Immunohistochemistry evaluation’ part of the revised manuscript. (Page 11, line 373-374)

how many samples/cells were analyzed?

Answer: Twenty samples in each five groups were evaluated with Ki-67 staining, and the same number of samples were evaluated with CD31 staining. (Page 11, line 370-372)

4.8.

Did authors prepare also negative and positive controls? This is essential for interpreting results in a good way.

Answer: (We have answered about the negative control and positive control in our study at the above answer to the comment for Fig. 4)

ln 319 - 320 - Which cells were counted?

In my opinion, this part is not clear. In the section Results, authors described generally the positive signal from cells in the tissue. I suggest correcting parts of section in Material and Methods and Results relating this topic.

Answer: We thank the reviewer for pointing this out. We examined the proportion of all TUNEL-positive cells including oocytes and granulosa cells in the examined ovarian tissue samples. Therefore, the following statement was a miswriting: “TUNEL-positive cells produced green fluorescence and follicles were considered as damaged when the oocyte nucleus and/or more than 10% of granulose cells were TUNEL-positive.” We have deleted the wrong part and have changed that statement as following: “TUNEL-positive cells produced green fluorescence and immunofluorescence images were obtained using a fluorescence microscope (Olympus, Japan) with 400x magnification.” (Page 11, line 394-397)

ln 321 - What was the magnification?

Answer: It was 400x, and we have indicated it in the last statement of that section and also have added the scale bar in the Figure 4. (Page 11, line 397)

4.9.

Did authors prepared negative controls?

Answer: (We have answered about the negative control and positive control in our study at the above answer to the comment for Fig. 4)

How many replicates were done?

Answer: Western blot was repeated five times both for caspase-3 and AMH analyses. (Page 12, line 418)

How was the analysis of signal was performed?

Answer: The signal (expression) of AMH was quantified using the ImageJ program and we have newly presented it as a graph in Figure 5C. We have added the following statement at the end of ‘4.9. Western blot’ section in the revised manuscript: “The expression of protein was quantified using the ImageJ program.” (Page 12, line 419)

4.11.

353 - Only follicle count was statistically analyzed?

Answer: We have corrected this part, as following: “The results of follicle counts, number of Ki-67 positive cells, proportion of CD31 positive area, and TUNEL-positive cells within the samples were compared with Student's independent t-test or analysis of variance (ANOVA) using SPSS version 12.0 software (SPSS Inc., USA). A P-value < 0.05 was considered statistically significant.” (Page 12, line 429-433)

Conclusion 

360 - 362 - Why authors only mention part of their results, when they e.g. in lns 189 -191 presenting differences in DNA frag and in ln 227 stating that they observed endocrine function restoration.

Answer: We thank the reviewer for pointing this out and we have changed that statement as following: “Although vitrification has shown promising results in recent experimental studies, our present study using a xenotransplantation model indicates that slow freezing method exhibits superior results as compared to vitrification in terms of primordial follicle preservation, vascularization, follicular cell proliferation, DNA damage, and AMH expression.” (Page 12, line 440-443)

362- I disagree with this statement in that form. In my opinion, it is presumptuous too much.

We thank the reviewer for pointing this out and we agree with the reviewer. Therefore, we have changed that statement as following: “Slow freezing should be still regarded as an effective option for ovarian tissue preservation for women with cancer who are interested in preserving their fertility, and further studies should be directed towards standardization of the vitrification method.” (Page 12, line 446-449)

We appreciate the thoroughness with which the reviewers and editors regarded our manuscript and hope that it is now suitable for publication in International Journal of Molecular Sciences.

Thank you for your consideration. We look forward to hearing from you soon.

 With best regards,

Tak Kim MD, PhD

Department of Obstetrics and Gynecology, Korea University College of Medicine, 126-1, 5-ga Anam-dong, Seongbuk-gu, Seoul 136-705, South Korea

Telephone: +82-2-920-5646

Fax: +82-2-921-5357

Reviewer 2 Report

This is an interesting manuscript from Lee et al., investigating the differential effect of slow-freezing (SF) versus vitrification (VT) approaches in maintaining ovarian tissue structure and function. Authors find that both methods severly affect tissue characteristics, with some lighter effects of SF. However, results presented do not testify only in favor of the exclusive tissue degradation as the major reason of trasplantation failure. In fact, lack of responsivity to pituitary hormones could partecipate to the observed issue degeneration in vivo. In fact, tissues express AHM after thowing but they loose this function after transplantation. Specifically:

1- Additional histochemical evaluation should be performed in pre-freezing and freshly thowed tissues (the latter from both SF and VT). In particular, immuno-staining for LH- and FSH- receptor should performed in order to address gonadotropins receptivity of transplanted tissue.

2- Tissues have been transplanted into adult ovariectomized animals, thus serum levels of LH and FSH before transplantation should be performed. Also estradiol should be quantified after transplantation in order to address tissue responsivity.

3-Possibly, in vitro stimulation with LH to detect estradiol relase should be performed.

4-TEM section of control (non freezed) tissue should be presented as reference.   

Author Response

RE: Manuscript ID. ijms-441579

Title: Comparison Between Slow Freezing and Vitrification for Human Ovarian Tissue Cryopreservation and Xenotransplantation

Thank you very much for your March 14 2019 decision letter. We appreciate the opportunity to revise and resubmit our manuscript. We have carefully read the reviewer’s constructive comments and have responded to them point-by-point below. And the revisions are highlighted using the "Track Changes" function in our revised manuscript file. We hope that our responses to the reviewer’s comments are adequate.

Thank you for your comments. We have taken into consideration your comments and revised the manuscript as follows.

Comments:

This is an interesting manuscript from Lee et al., investigating the differential effect of slow-freezing (SF) versus vitrification (VT) approaches in maintaining ovarian tissue structure and function. Authors find that both methods severly affect tissue characteristics, with some lighter effects of SF. However, results presented do not testify only in favor of the exclusive tissue degradation as the major reason of trasplantation failure. In fact, lack of responsivity to pituitary hormones could partecipate to the observed issue degeneration in vivo. In fact, tissues express AHM after thowing but they loose this function after transplantation. Specifically:

1- Additional histochemical evaluation should be performed in pre-freezing and freshly thowed tissues (the latter from both SF and VT). In particular, immuno-staining for LH- and FSH- receptor should performed in order to address gonadotropins receptivity of transplanted tissue.

Answer: I appreciate for your kind suggestions. Authors agree with reviewer’s opinion, however, the purpose of this study was to compare the effectiveness and safety of the two methods for ovarian tissue cryopreservation and xenotransplantation. Many mechanisms such as apoptosis, hormonal effect, ischemic damage, or necrosis may be influenced tissue damage during the cryopreservation and transplantation. At present, authors are evaluating those mechanisms including hormones as reviewer’s suggestions.

2- Tissues have been transplanted into adult ovariectomized animals, thus serum levels of LH and FSH before transplantation should be performed. Also estradiol should be quantified after transplantation in order to address tissue responsivity.

Answer: Experiment was performed using 5-6 weeks old female severe combined immunodeficiency mouse (weight: 16-20g). The total blood volume of mouse is about 1.5-2.0ml. Technically, LH and FSH measurement needs about more than 1-1.5 ml which is critical for mouse. Thus LH and FSH measurement before transplantation is not available. In order to obtain the exact results without dilution, only one or maximum two hormones can be measured after mouse sacrifice. Authors evaluated FSH and Estradiol levels in our other experiment and we will submit out data in near future. Thank you for your kind suggestions.

3-Possibly, in vitro stimulation with LH to detect estradiol relase should be performed.

Answer: Thank you for your comment and suggestions. In fact, in vitro ovarian tissue culture is another big issue due to the difficulty of oocyte proliferation. Some researchers are trying to 3D ovarian tissue culture or artificial ovary to get the in vitro oocyte proliferation without transplantation. Definitely authors will check the estradiol release with LH stimulation using xenotransplantation model in next experiment.

4-TEM section of control (non freezed) tissue should be presented as reference.

Answer: Thank you for your comment and suggestions. TEM morphological evaluation of primordial follicles was performed in three samples from each group: SF and VT group. And we have found that the primordial follicles in VT group were deformed and destructed as we presented in the Figure 6, whereas the primordial follicles in SF groups were morphologically intact. As the reviewer pointed out, the control is missing in this images. However, we performed TEM examinations to compare the morphological findings of follicles between SF and VT group, and the differences between them might be clearly identified in the Figure 6.

We appreciate the thoroughness with which the reviewers and editors regarded our manuscript and hope that it is now suitable for publication in International Journal of Molecular Sciences.

Thank you for your consideration. We look forward to hearing from you soon.

With best regards,

Tak Kim MD, PhD

Department of Obstetrics and Gynecology, Korea University College of Medicine, 126-1, 5-ga Anam-dong, Seongbuk-gu, Seoul 136-705, South Korea

Telephone: +82-2-920-5646

Fax: +82-2-921-5357

Round  2

Reviewer 2 Report

Authors have not addressed one single criticism arisen in the previous review. I can't say I'm satisfied with this revision.

Author Response

Manuscript ID. ijms-441579

Title: Comparison Between Slow Freezing and Vitrification for Human Ovarian Tissue Cryopreservation and Xenotransplantation

We appreciate the opportunity to revise and resubmit our manuscript. We have carefully read the reviewer’s constructive comments and have responded to them point-by-point below. And the revisions are highlighted using the "Track Changes" function in our revised manuscript file. We hope that our responses to the reviewer’s comments are adequate.

Dear reviewer, 

Thank you for your comments. Authors apologize that answers of the first revision might disappoint the reviewer. According to the reviewer’s recommendations, further examinations were performed including the immunofluorescence staining for luteinizing hormone (LH) receptor and follicle stimulating hormone (FSH) receptor in the ovarian tissues to assess the gonadotropins receptivity. Interestingly, the proportion of FSHR-positive area and LHR-positive area were higher in the transplanted tissue group (SF-T and VT-T) than those in the freezing group (SF and VT). Those results implicated that the freezing process might induce the reduction of gonadotropins receptivity of human ovarian tissues, but those were restored in some degree after transplantation. Among the engrafted ovarian tissues, the proportion of the area positive for gonadotropin receptors was higher in the slow freezing group (SF-T) than vitrification group (VT-T). This results indicated that slow freezing protocol might be better option in terms of restoration of gonadotropin receptivity in human ovarian tissues after the cryopreservation and transplantation. We appreciate that this result much improved the quality of our manuscript.

In addition, the analyzed results of immunohistochemistry for Ki-67 and CD31 were added in the pre-freezing tissues (control group) and the freshly thawed tissues from both SF and VT groups (figure 3 was modified). Also, authors have added the figure of transmission electron microscopy (TEM) section of the ovarian tissue from control group as reference (Figure 6-C). The follicle from vitrification group was deformed and destructed by ice crystal formation after thawing, whereas the follicle from the control group and the slow freezing group were morphologically intact.

On the other hand, the authors could not check the serum level of LH and FSH in the SCID mice on which the ovarian tissues were transplanted because of the reason that the authors had explained at the answer for the first revision: Technically, LH and FSH measurement needs about 1-1.5 ml of whole blood to obtain appropriate amount of serum, which is critical for mouse. Thus LH and FSH measurement before transplantation is not available. In order to analyze the exact results without dilution, only one or maximum two hormones can be measured after mouse sacrifice.

We appreciate the thoroughness with which the reviewers and editors regarded our manuscript and hope that it is now suitable for publication in International Journal of Molecular Sciences.

Thank you for your consideration. We look forward to hearing from you soon.

Best regards,

Tak Kim MD, PhD

Department of Obstetrics and Gynecology, Korea University College of Medicine, 126-1, 5-ga Anam-dong, Seongbuk-gu, Seoul 136-705, South Korea

Telephone: +82-2-920-5646

Fax: +82-2-921-5357
